# Predicting the Diagnosis of Prostate Cancer with a Novel Blood-Based Biomarker: Comparison of Its Performance with Prostate-Specific Antigen

**DOI:** 10.3390/cancers16152619

**Published:** 2024-07-23

**Authors:** Johnmesha L. Sanders, Kenneth A. Iczkowski, Girish V. Shah

**Affiliations:** 1Pharmacology, College of Pharmacy, University of Louisiana at Monroe, Monroe, LA 71209, USA; sanderjl@warhawks.ulm.edu; 2Department of Pathology and Laboratory Medicine, School of Medicine, University of California—Davis, Sacramento, CA 95817, USA; kaiczkowski@ucdavis.edu

**Keywords:** novel blood-based biomarker, clinical studies, validation of a biomarker, prostate cancer

## Abstract

**Simple Summary:**

For decades, a blood test showing elevated prostate-specific antigen (PSA) has been the mainstay for detecting prostate cancer. However, the PSA is often elevated in the absence of cancer. Here, we show that the neuroendocrine marker (NEM) outperforms the PSA test, both for sensitivity and for the exclusion of false-positive results.

**Abstract:**

The purpose of this study was to assess the efficacy, specificity, and predictive value of a newly discovered biomarker, Zinc finger-like1 protein (referred to as neuroendocrine marker, NEM) for the detection of prostate cancer (PCa). We retrospectively analyzed banked plasma samples from 508 men, with a median age of 67 years (range 48–97), to compare the performance of NEM and PSA in predicting subsequent histologic PCa. The cohort consisted of four groups of patients visiting a urology clinic: (1) patients not diagnosed with either benign prostatic disease or prostate cancer (PCa) were defined as normal; (2) patients diagnosed with benign hyperplasia (BPH) but not PCa; (3) patients with confirmed PCa; and (4) patients with cancer other than PCa. The normal men displayed a mean NEM plasma level of 0.948 ± 0.051 ng/mL, which increased to 1.813 ± 0.315 ng/mL in men with BPH, 86.49 ± 15.51 ng/mL in men with PCa, and 10.47 ± 1.029 ng/mL in men with other Ca. The corresponding concentrations of prostate-specific antigen (PSA) in these subjects were 1.787 ± 0.135, 5.405 ± 0.699, 35.77 ± 11.48 ng/mL, and 8.036 ± 0.518, respectively. Receiver operating characteristic (ROC) curve analysis was performed to compare NEM and PSA performance, and the Jouden Index for each biomarker was calculated to determine cut-off points for each biomarker. The area under the ROC curve to predict PCa was 0.99 for NEM and 0.81 for PSA (*p* < 0.0001). The cut-off for NEM was at 1.9 ng/mL, with sensitivity of 98% and specificity of 97%. The corresponding PSA values were 4.4 ng/mL, with sensitivity of 76% and specificity of 95%. The predictive value of each biomarker in a patient was matched with his pathologic data to determine the accuracy of each biomarker. NEM was more accurate than PSA in differentiating cancer from benign conditions, such as BPH or prostatitis. In conclusion, NEM was a better predictor of PCa than PSA in patients visiting urology clinics. NEM tests, either alone or in conjunction with other biomarkers, provide a reliable, non-invasive, and inexpensive test to remarkably reduce false positives and thereby reduce the number of diagnostic biopsies and associated painful procedures and the loss of quality of life.

## 1. Introduction

Prostate cancer (PCa) is one of the leading diseases affecting elderly men [1]. The early detection of PCa is important to successfully treat the patients as there is no effective treatment for advanced PCa. Although digital rectal examination (DRE) has previously been used in the diagnosis of PCa, its accuracy is low [2]. With the emergence of prostate-specific antigen (PSA) as a serum-based PCa biomarker, it was coupled with DRE to improve the detection of PCa [3]. Unfortunately, this has led to the detection of many cases of PCa that were either false positives or low-risk PCa that did not require immediate treatment [4]. Several modifications have been made to the PSA test to improve the specificity, such as the measurement of PSA velocity, free-to-total PSA ratio, PSA density (PSAD), and 4K test [5]. The major limitation of PSA or PSA-derived tests is that PSA is a kallikrein protease and is a normal secretory product of luminal cells of the prostate gland [6]. Although the production of PSA increases in cancer, the increase also occurs in several benign conditions as well, including benign prostatic hyperplasia (BPH), prostatitis, or inflammatory conditions of the prostate [7,8,9,10,11]. Therefore, screening for PCa using the PSA test has a significant risk of overdiagnosis. More than one million men undergo prostate biopsy each year in the USA, with the majority showing either no PCa or low-risk cancer that is unlikely to impact quantity or quality of life [12]. Indeed, other modern methods of PCa diagnosis such as prebiopsy diagnostic MRI are emerging to evaluate potential prostate pathology, but they are not suitable for screening a large number of cases [13]. Therefore, the application of newly discovered serum-based cancer-specific biomarkers is critical to improve the detection of PCa in at-risk populations and, thereby, reduce unnecessary biopsies.

Recent studies have reported that zinc finger protein-like 1 (ZFPL1) is selectively secreted by malignant, but not benign, prostate epithelial cells [14]. Moreover, ZFPL1 is co-localized with neuroendocrine markers such as chromogranin A in the tumor cells of the prostate and is packaged in exosomes and secreted in circulation. Since the originating cells of ZFPL1 in the prostate are neuroendocrine cells, we refer to ZFPL1 in the prostate as neuroendocrine marker (NEM) in this study. We observed a many-folds increase in its mRNA levels in a malignant prostate accompanied with a many-folds increase in NEM protein levels in the sera of patients with PCa, as compared to the age-matched normal male population [14,15]. In addition, we developed a unique monoclonal antibody against ZFPL1 and used it to develop an immunosensor-based rapid assay for the measurement of ZFPL1 that is quick, at least 50–100X sensitive as ELISA, and can be used for the concurrent measurement of a large number of clinical samples [15,16]. Here, we report the comparative results of plasma PSA and ZFPL1 levels to accurately predict PCa from a retrospective cohort of 508 patients.

## 2. Methods

*Patients.* We retrospectively collected data and plasma samples from 508 men from plasma banks of following three sources: (1) Medical College of Wisconsin (MCW; a total of 347 samples); (2) Louisiana State University Medical Center at Shreveport (LSUHSC-S; a total of 81 PCa samples); and (3) 80 PCa samples from Individumed Services (Frederick, MD). The subjects of the cohort included men in the 48–97 age range who visited the clinics for urological issues (please see Table 1 for details). The patient populations of MCW included (i) benign or normal patients; (ii) those with BPH but not PCa; (iii) those with pathologically confirmed PCa; and (iv) patients with other cancers (but not PCa). Benign or normal patients were those who visited MCW urology clinic and had a PSA test before a biopsy or a transurethral resection that had confirmatory benign results and no cancer on follow-up.

The clinical information of the patients did not identify the individuals but provided clinical diagnosis, serum PSA levels at the time of sample collection, histology, Gleason score, and TNM stage if a prostatectomy was performed. However, some institutions provided more complete clinical information that included tumor stage and histologic data. Since the primary objective of our study was to test the predictive value of NEM to diagnose PCa accurately and compare it with that of PSA, plasma samples and clinical diagnosis with/without complete histopathology were sufficient for inclusion in our study. The protocols for the acquisition and assay of human plasma samples were approved by the Institutional Review Board of the University of Louisiana at Monroe (ULM clinical study protocol 001, 20 May 2019) and MCW (PRO16747). Written consent was provided by patients at the time of tissue collection to the clinical teams. Because the samples for this study were archived and banked at the referred institutions, the study fell under Exempt 4 category under federal guidelines.

*Measurement of Biomarker levels in plasma of human subjects.* The samples were used to determine NEM and PSA concentrations by immunosensor assay, as described previously [15,16]. The assay is linear over a range of 1–64 pg with a sensitivity of 1 pg/50 μL. This enables the accurate measurement of serum NEM levels in as little as 0.1 μL serum. The assay has been examined for its accuracy, precision, recovery, and linearity. The dilution curve of human serum was parallel to the NEM standard curve in the range of 0.1–2 μL serum [15]. Negative controls are the serum pool from patients who have undergone prostatectomy (serum PSA < 0.003 ng/mL), and the positive controls are the serum pool of patients with PCa (confirmed by biopsy). Intra-assay variations were <5% over the course. We monitored inter-assay variation in 33 assays in the present study by running the same pool sample in every assay. The inter-assay variation was 14.4%.

*Statistical analysis*. The results were analyzed by descriptive statistics. Receiver operating characteristic (ROC) curves were plotted as sensitivity vs. 1-specificity for NEM and PSA. The NEM and PSA area under the curve (AUC) were used to determine overall performance of predicting PCa vs. no cancer. Youden’s index (sensitivity + specificity − 1) was determined to obtain optimum cut-off point for each biomarker to be used as a predictor of PCa. The accuracy of prediction based on each biomarker level in a patient’s plasma was verified by matching it with the patient’s clinicopathologic data. Then, the numbers of true positives (TP), true negatives (TN), false positives (FP), and false negatives (FN) were determined. The accuracy, positive predictive value (PPV), and negative predictive value (NPV) for each biomarker were calculated. Student’s t-test was used for the comparison. Additionally, the relationship between plasma NEM and PSA levels with tumor progression, as indicated by tumor stage (International Society of Urological Pathology or ISUP), was evaluated. All tests were two-tailed, and significance was defined as *p* < 0.05. Analyses were performed and graphed using Prism 8 software (GraphPad Software, Inc., Boston, MA, USA).

## 3. Results

*Cohort*: Our cohort consisted of 508 men with urologic issues in an age group of 48–97. The plasma samples of these patients were banked in biorepositories of the respective institutions. Their median PSA was 7.05 ng/mL. PCa was pathologically confirmed in 311 men. The remaining subjects included men without cancer (referred to as normal), those with benign prostatic hyperplasia (BPH), with cancers other than PCa. Table 1A summarizes the clinical profiles of the men. The PCa patients from LSUHSC and Individumed Services were further sub-categorized based on tumor stage because we had complete clinical data for each patient (Table 1B,C). The archived plasmas were sent to our laboratory for NEM and total PSA assays, as described in the Methods section.

*Data Analysis*: We first examined whether there were significant differences in the age of the patients in different groups based on clinical status, as indicated in Table 1C. For this, we performed descriptive statistics and examined significant differences between the age range of the groups via Chi-square test as well as one-way ANOVA. We did not observe significant differences between the age ranges of different groups.

Next, we determined the plasma levels of NEM and PSA in individual samples. The biomarker values of all samples were appropriately distributed in four categories: N (normal), BPH, PCa, and other Ca. The values were then plotted in box-and-whisker plots for NEM (Figure 1A) and PSA (Figure 1B), where the line in the box presents the median value for each group. The data were then analyzed using Kruskal–Wallis test using Chi-square (df:3) distribution. The results reveal that median NEM values in normal individuals were 0.85 ng/mL, which increased to 1.53 in BPH, increased even further to 46.47 ng/mL in the PCa group, and were moderately higher than normal but remarkably lower than PCa at 6.4 ng/mL in the other Ca group. Comparatively, median plasma PSA levels were 1.37 ng/mL in normal, 4.62 ng/mL in BPH, 8.38 ng/mL in PCa, and 7.28 ng/mL in the other Ca group. The Kruskal–Wallis test showed significant differences in the ranks of values in all four groups. Moreover, one-way ANOVA multiple comparison test also revealed that mean plasma NEM level ranks in the PCa group were significantly different from those of the normal, BPH, and the other Ca groups (*p* < 0.05). Similarly, the Kruskal–Wallis test of PSA levels identified significant differences between the ranks of controls and PCa groups and controls and the other Ca groups (*p* < 0.05). However, one-way ANOVA multiple comparison analysis did not find significant differences in PSA among these groups.

*Receiver Operator Curves for NEM and PSA*: Next, we examined the performance of the two biomarkers via receiver operating characteristic curves (ROCs). The ROC curve is a plot of the true-positive rate (sensitivity) as a function of the false-positive rate (1-specificity) for different cut-off points of a biomarker. The ROC curve is a fundamental tool for diagnostic test evaluation [17,18] and to compare the diagnostic performance of two or more laboratory or diagnostic tests [19]. The results in Figure 2 show that in a cohort of 98 normal subjects and 311 PCa patients, the AUC of the NEM ROC curve (black) was 0.9935, which is very close to a perfect 1.0. The AUC of the PSA ROC curve (blue) with the same cohort was 0.8140, demonstrating that the NEM was a better PCa biomarker than PSA. Each point on the ROC curve represents a sensitivity/specificity pair, corresponding to a particular decision threshold. The area under the ROC curve (AUC) is a measure of how well a biomarker can distinguish between two diagnostic groups (diseased/normal). We compared the ROC characteristics of NEM and PSA curves by z statistics. The NEM test displayed significantly better performance than PSA (*p* < 0.0001). We then calculated cut-off points for both biomarkers by calculating the Jouden Index from their respective ROC curves. The highest J value for NEM was 0.95, which translated to a cut-off value of 1.972 ng/mL at a sensitivity of 98.07% and specificity of 96.94%. The corresponding J value for PSA was 0.6607 and a cut-off value of 4.26 ng/mL with sensitivity of 76.21% and specificity of 94.9%.

*Comparison of Biomarker Performance*: Once we determined the cut-off point for each biomarker, we then relooked at the biomarker levels of each of the subjects and matched with their clinical diagnosis to determine the accuracy of prediction and calculate positive predictive and negative predictive values. These results are presented in Table 2A–D.

The results in Table 2D show that NEM performed better than PSA. Specifically, NEM was markedly more accurate in detecting true negatives compared to PSA.

*NEM and tumor stage*: Currently, there is no blood-based biomarker that can reliably predict the tumor progression of PCa. This is still performed using invasive procedures such as needle biopsies followed by histologic evaluation [2]. We examined whether plasma NEM levels significantly rise with an increase in tumor stage. The results in Figure 3A,B show that plasma NEM levels rose with increasing tumor stage (Figure 3A), but plasma PSA levels did not (Figure 3B).

The objective of Figure 4 was to examine if the biomarkers PSA and NEM accurately discriminate the cases of cancer from benign ones on the basis of the cut-off levels determined in Figure 2. We plotted scattergrams of all BPH cases for NEM and PSA (Figure 4A) and all cases (including BPH and PCa) that showed PSA levels between 4 and 10 ng/mL (Figure 4B). The horizontal lines drawn on the plot indicate the cut-off levels of a respective biomarker. The results in Figure 4A show that only 6 of 19 BPH cases were above the cut-off line of NEM, and, eventually, four of them were among those cases that were PCa-negative when initially detected but later developed PCa (3–10 years after initial testing). In contrast, all BPH cases showed PSA levels that were higher than the cut-off value, as determined by the ROC curve. Figure 4B shows a similar phenomenon in BPH and/or PCa cases where plasma PSA levels were between 4 and 10 ng/mL. Here, again, NEM could differentiate benign cases from the PCa ones (Figure 4B).

## 4. Discussion

Although the PSA test is widely used for screening men for PCa, there is much concern about its low predictive value, high number of false positives, and the detection of clinically insignificant PCa [2,7,20,21,22]. Recently, several new tests of PSA derivatives such as PHI and 4K tests have improved the predictability of cancer detection, but there is still a concern about false positives and insignificant prostate cancer that does not require aggressive treatment [23,24,25]. This is because PSA or its derivatives are normal secretory products of the prostate, and their secretion increases with the increase in prostate size rather than the PCa [21]. This limitation of PSA can be eliminated by identifying a biomarker that is selectively secreted by a malignant, but not benign, prostate. A recent report from this laboratory demonstrated that NEM expression in prostate cells is cancer-specific, and plasmas of PCa patients show a many-folds increase in circulating NEM levels [14].

The present study conducted systematic determination of plasma NEM and PSA levels in a cohort consisting of normal, BPH, PCa, and other Ca patients. The statistical analysis of these data led to the following observations. First, NEM could distinctly discriminate PCa patients from all other groups. Specifically, the results of BPH patients were differentiated into two groups, those closer to normal NEM levels and those greater than the cut-off point for PCa. We were highly encouraged to report that at least four BPH cases showed elevated NEM levels but were classified as normal by the PSA test as well as pathological tests at the time of sample collection. However, they were diagnosed as PCa-positive 3–10 years after the sample collection. This suggests that NEM may detect PCa much earlier than PSA or pathological tests. Although this is an observation from a small number of subjects in a study that was not designed to follow individual PCa patients over a period of time, this preliminary observation provides a strong justification to undertake a properly designed larger study to address this important issue.

A major deficiency of the PSA test is that its serum levels may rise to 10 ng/mL in PCa as well as other prostate diseases, such as BPH or prostatitis [20,26,27]. Therefore, serum PSA levels between 4 and 10 ng/mL are considered the gray zone, and these patients need to go through highly invasive procedures such as biopsy to confirm the PSA test. The present results suggest that NEM may help identify true-negative and true-positive PCa cases in the gray-zone PSA population. To summarize, NEM performed better than PSA in ROC analysis as well as other biomarker characteristics. Notably, the NPV of NEM was remarkably better than that of PSA. This is to be expected since NEM is selectively secreted by malignant, but not benign, prostate cells, and is expected to reduce false positives [14,15]. Overall, the results suggest that NEM is more reliable than PSA in predicting PCa and may help in identifying potential PCa cases among gray-zone PSA patients in a non-invasive manner. This improvement alone can significantly reduce the number of diagnostic biopsies and provide relief from the pain and expense of biopsy to patients.

Next, we examined whether plasma NEM levels increase with tumor progression. Since NEM is secreted by the prostate tumor cells that express neuroendocrine genes, it is conceivable that the NEM test will provide a measure of secretory activity of the neuroendocrine cell populations in the tumor [15]. Immunohistochemistry (IHC) of primary prostate tumors revealed that 47–100% of PCa demonstrates foci of neuroendocrine differentiation [28,29,30,31,32,33]. PCa patients with “low or normal” serum PSA levels usually display higher neuroendocrine (NE) secretions and aggressive growth, suggesting that the genes associated with neuroendocrine features may help detect aggressive PCa [34,35,36]. Based on these findings, it is conceivable that the NEM test will detect PCa cases with “normal” PSA levels and that serum NEM levels will increase with an increase in the NE cell population. The present results, showing that plasma NEM levels increased with an increase in tumor stage, support the possibility that NEM may serve as a prognostic marker for PCa. Additional studies will examine this possibility further.

This is the first clinical study evaluating the performance of NEM vs. PSA. Therefore, it is important to mention the limitations of this study. The first limitation is the presence of a predominantly Caucasian population in our cohort, as compared to African Americans. Therefore, we could not examine racial differences in plasma NEM levels. Second, this was an observational retrospective study lacking follow-up data on individual patients’ plasma PSA levels and other clinicopathologic parameters over a period. Therefore, the post-treatment kinetics of NEM secretion remains to be examined. Third, the absence of prospective data in this study raises the possibility that NEM may or may not be as accurate in predicting PCa as observed in the present study. Fourth, we examined the performance of biomarkers and determined the cut-off points only for the presence/absence of PCa. Therefore, this cut-off point may not be applicable for the detection of clinically significant prostate cancer. In summary, NEM was a significantly better predictor than PSA alone for PCa in males. We propose that the NEM test, with or without PSA, is a simple, inexpensive tool that can be used to diagnose PCa as well as to reduce the problems of the PSA test that includes the overdiagnosis of PCa and false positives and, consequently, the morbidity of unnecessary biopsies.

## Figures and Tables

**Figure 1 cancers-16-02619-f001:**
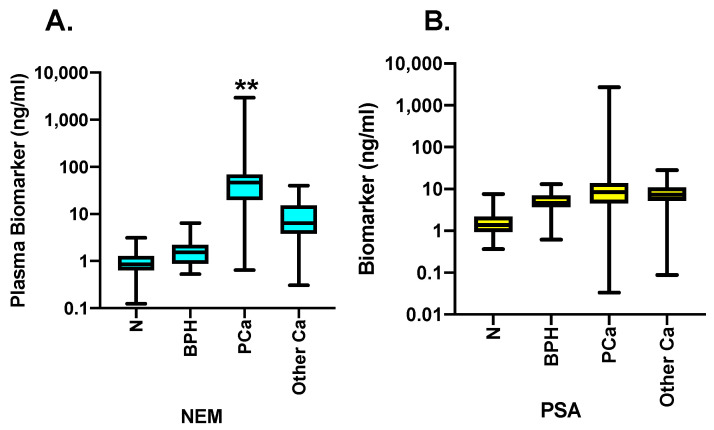
Plasma NEM and PSA levels in normals, BPH, PCa and other Ca groups. Box-and-whisker plots for NEM and PSA that show the summary and distribution of plasma NEM and PSA levels in our cohort of 508 patients that was divided into four groups: N = normal, (*n* = 98), BPH (*n* = 19), PCa (*n* = 311), and other Ca (*n* = 80). The data were statistically analyzed by Kruskal–Wallis test to examine the significance among ranks of all four groups. Both biomarkers displayed significant differences between PCa group and all other groups. The data were also analyzed by one-way ANOVA multiple comparison test. One-way ANOVA showed significant differences between NEM levels of PCa group and other three groups. However, no significant differences were found in plasma PSA levels between these groups. ** *p* < 0.05.

**Figure 2 cancers-16-02619-f002:**
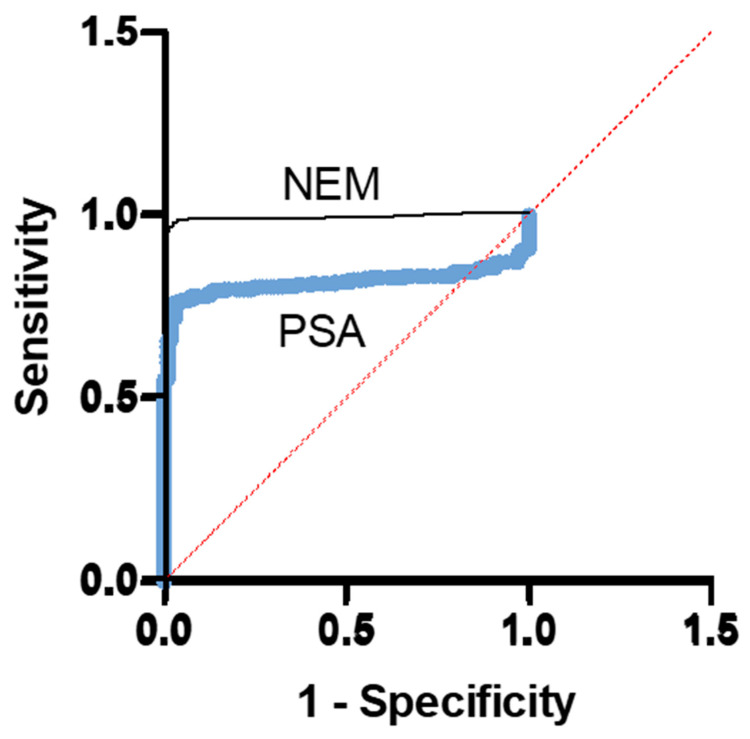
Receiver operating curve (ROC) analysis of NEM and PSA. The curves were derived by Prism (8.0) after entering the data of plasma levels of both biomarkers of the cohort consisting of normal (*n* = 98) and PCa (*n* = 311) subjects. The area under the curve (AUC) for NEM (black) was 0.9935 ± 0.0034 (*p* < 0.0001). The AUC for the PSA (blue) was 0.8140 ± 0.0209 (*p* < 0.0001).

**Figure 3 cancers-16-02619-f003:**
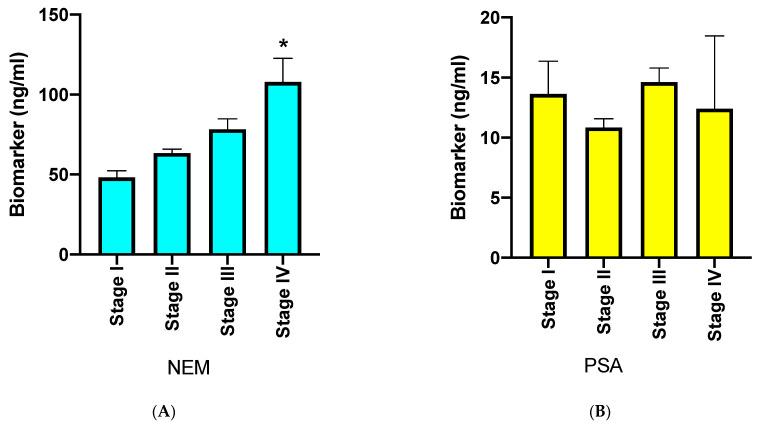
Bar graphs of plasma NEM and PSA levels at different stages of PCa. (**A**): A bar graph showing mean ± SEM of plasma NEM levels according to AJCC prognostic stage group. The cohort consisted of stage I (*n* = 100); stage II (*n* = 89); stage III (*n* = 57); and stage IV (*n* = 4). (**B**): A bar graph showing mean ± SEM of plasma PSA levels as a function of stage. The cohort consisted of stage I (*n* = 100); stage II (*n* = 89); stage III (*n* = 57); and stage IV (*n* = 4). * *p* < 0.05 between Stage I and Stage IV (One Way ANOVA multiple comparison).

**Figure 4 cancers-16-02619-f004:**
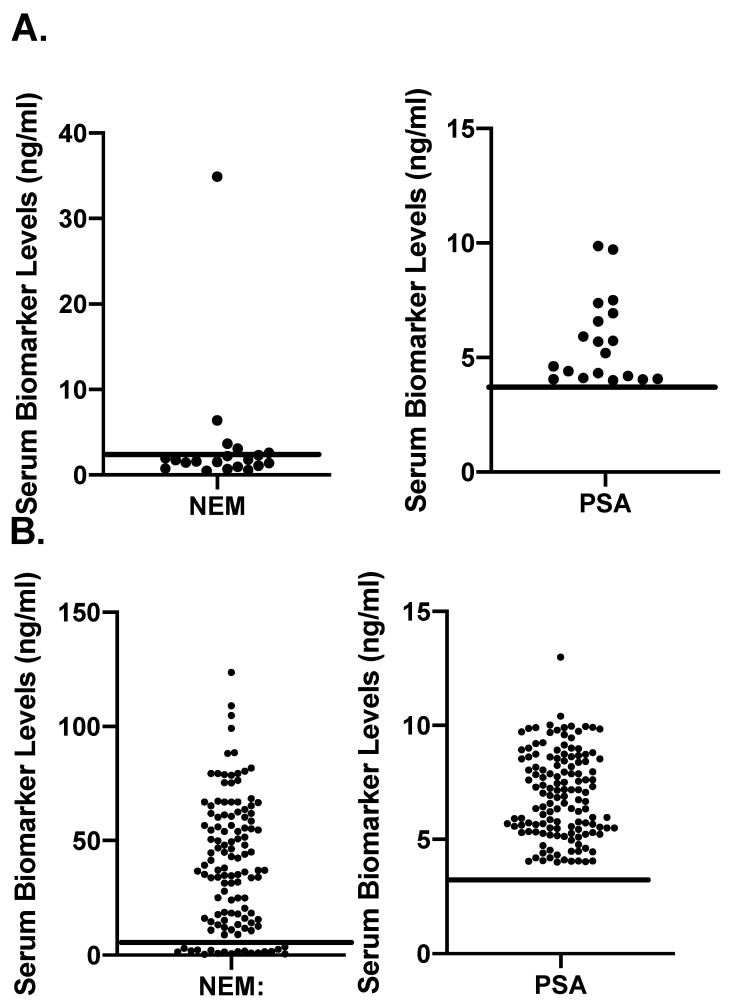
Scattergrams of plasma NEM and PSA levels in cohort of patients showing PSA levels between 4 and 10 ng. (**A**): Scattergrams of plasma NEM and PSA levels in patients diagnosed as BPH (*n* = 19); (**B**): scattergrams of plasma NEM and PSA levels in patients displaying plasma PSA levels in the range of 4–10 ng/mL, irrespective of their clinical status (*n* = 135).

**Table 1 cancers-16-02619-t001:** Clinical profile and sources of the cohort.

A: The cohort *					
Source	Normal	BPH	PCa	Other Cancers	Total
MCW	98	19	150	80	347
LSUHSC	0	0	81	0	81
Individumed	0	0	80	0	80
Total	98	19	311	80	508
B: Racial distribution					
	Normal	BPH	PCa	Other Cancers	Total
African Americans	6	1	54	3	64
Caucasian	92	18	257	77	444
Total	98	19	311	80	508
C: Clinical profile of the cohort					
Number of Subjects (n), Source		Average Age (years)	Age Range (years)	Clinical Diagnosis	
98 (MCW)		68.66	58–86	Normal	
19 (MCW)		71.47	59–89	BPH	
311 (MCW, LSUHSC, Individumed)		67.01	59–89	PCa	
80 (MCW)		71.3	54–93	Other Cancers	
D: AJCC Prognostic stage distribution of PCa patients					
Number of Subjects (n)		Average Age (years)	Age Range (years)	Tumor Stage	
11 (LSUHSC, Individumed)		60.82	48–71	Stage I	
89 (LSU, Individumed)		61.16	48–75	Stage II	
57 (LSUHSC, Individumed)		62.26	48–75	Stage III	
4 (LSUHSC)		59.67	54–62	Stage IV	

* There were no statistically significant differences in age among all four groups (descriptive statistics and one-way ANOVA).

**Table 2 cancers-16-02619-t002:** Biomarker performance comparison. Predictive accuracy of a biomarker in clinical conditions.

A: Normal Patients					
Biomarker	FP	FN	TP	TN	Total
NEM	3	0	0	95	98
PSA	8	0	0	90	98
B: BPH Patients					
Biomarker		PCa+	PCa−	Total	
NEM		6	13	19	
PSA		14	5	19	
C: PCa Patients					
Biomarker		PCa+	PCa−	Total	
NEM		310	1	311	
PSA		269	42	311	
D: Biomarker Performance.					
Biomarker		Accuracy	PPV	NPV	Subjects (n)
NEM (normal + PCa)		0.933	0.982	0.993	311 + 98
PSA (normal + PCa)		0.880	0.965	0.719	311 + 98
NEM (BPH + PCa)		0.981	0.925	0.981	311 + 19
PSA (BPH + PCa)		0.821	0.821	0.331	311 + 19

## Data Availability

The data presented in this study are available in this article.

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
