# Peer review of "Predicting the Diagnosis of Prostate Cancer with a Novel Blood-Based Biomarker: Comparison of Its Performance with Prostate-Specific Antigen"

_cancers, 2024, doi:10.3390/cancers16152619_

Round 1

Reviewer 1 Report

Comments and Suggestions for Authors

It was a clinical study about introducing a new biomarker, Zinc finger-like1 protein (NEM), for the detection of prostate cancer and comparing its effectiveness with PSA, as the most prevalent biomarker used for the detection of prostate cancer. Here are some comments on this study that should be considered before publication:

1.     Please correct Table 3.

2.    Please check the references again, some of them have missing parts that need to be completed. 

Comments on the Quality of English Language

 Please check the whole text again, there are some tiny mistakes in the text that should be corrected.

Author Response

We greatly appreciate Reviewer 1 for the review of our Manuscript. We have addressed the issues raised by correcting tale 3 (Please see Table 2) in revised MS. We have also carefully edited  the MS to correct miscues. Thanks again

Reviewer 2 Report

Comments and Suggestions for Authors

The authors have developed a quantitative assay based on a proprietary antibody  and device to quantify the biomarker ZFPL1 in plasma/serum to diagnose the presence of prostate cancer.

Non-invasive methods to improve the screening of prostate cancer based on PSA would have a relevant impact on the clinical management of patients and is an important research topic.

Major revision

Manuscript:

NEM was also detected in small quantities in the brain and some peripheral neuroendocrine organs such as pancreas and ovary[14]. However, we observed many-folds increase in its mRNA levels in malignant prostate accompanied with many-folds increase in NEM protein levels in sera of patients with PCa as compared to the age-matched normal male population[14].”

 1.      Comment: Add space (in yellow)

 Manuscript:

“In addition, we have developed a unique monoclonal antibody against ZFPL1, and used it to develop an immunosensor-based rapid assay for the measurement of ZFPL1 that is quick, at least 100X sensitive as ELISA, and can be used for concurrent measurement of a large number of clinical samples [15].”

 2.      Comment: The reference to this assertion is an abstract where no direct comparison to an ELISA is performed. The author should specify what they mean with “at least 100X sensitive as ELISA” and adapt the claim.

 Manuscript:

“The clinical information of the patients did not identify the individuals, but provided TNM stage, histology type, tumor grade, serum PSA levels at the time of sample collection and Gleason

score for each sample if available.”

3.     3. Comment: The authors should specify at which time points of the disease course the sample were collected. For prostate cancer patients:  prior to biopsy, or prior to prostatectomy? During treatment? Which treatment?  For BPH patients: before, during or after treatment? For BCR patients: before prostatectomy? During BCR?

Manuscript:

“The assay is linear over the range of 1-64 pg with the sensitivity of 1 pg/50 μl. This enables the accurate measurement serum NEM levels in as little as 0.1 μl serum. The assay has been examined for its accuracy, precision, recovery, and linearity. The dilution curve of human serum was parallel to the NEM standard curve in the range of 0.1-2 μl serum. Negative controls are the serum pool from patients who have undergone prostatectomy (serum PSA <0.003 ng/ml); and the positive controls are the serum pool of patients with PCa (confirmed by biopsy). At present, intra, and inter-assay variations of the assay are < 5% and 9%, respectively.

 4.      Comment: The authors should show these data or add the reference to a published manuscript.

 5.      Comment: Table 1A: age should show min-max, average and statistical comparison among the 4 groups.

Table 1A and B: In table 1A PCA patients are 308, table 1B shows in total 145 patients with stage. Please explain the difference.

Table1: authors should add the total n for each table A,B and C, for example the total for 1A is 503. They should also add a column with all measurable used in the analysis per group (for example PSA).

Then the authors write:

“We first performed multiple linear regression analysis to evaluate for a possible relationship between plasma NEM levels with the age of the patient, presence of PCa, and PSA levels. In a population of 407 subjects”

Why 407? Why not 503?

Table 1C: were the samples collected again during BCR? Or was the analysis performed prior to treatment?

 6.      Comment: Figure 1 is unclear. Number and groups of patients analysed is not clear. Authors should have shown the Pearson correlation with each single variable. What is the correlation to Cancer? Yes/no value? What is the purpose to correlate a continuos variable to a binary variable? It would have been much more valuable to show and compare the distribution of NEM in the 4 patient groups (if samples were taken at the same time point), for example by kruskal wallis test and box plot.

 7.      Comment: Table 2A is missing something. Are the first 3 columns based on NEM and the second 3 on PSA as cut-off? This is missing. The cut off values in the text are different from those in the table anyway (1.8 instead of 1.787 and 4.26 and 4.216). These data would be best understood graphically with a box plot. The number of patients in the table do not correspond to table 1. If some patients have partial data they should be removed from the analysis and the different groups should be consistent in size. Same comments for Table 2B and C.

Manuscript:

Comparison of NEM and PSA for the prediction of PCa: Using 1.8 ng/ml as a cut-off for NEM, we assigned each patient’s status and matched it with his clinicopathologic data (Table 3). Only 2 of 97 normal patients displayed NEM levels greater than the cut-off value and were considered as False Positive (FP) (Table 3A). Other 95 showed NEM levels below 1.8 ng/ml and were considered true negative (TN). The PSA levels in the same cohort had 90 patients below 4 ng/ml and were considered TN, but 7 patients had PSA greater than 4 ng/ml and were considered false positive (FP).

 8.      Comment:  I would suggest to use the data to generate a ROC curve and then detect the cut-off at a specific sensitivity.

Manuscript:

“Please note that their plasma samples were collected several years earlier and were diagnosed PCa-negative based on serum PSA levels as well as clinicopathologic parameters. In contrast, only 5 of 18 BPH patients displayed plasma PSA levels lower than the cut-off limit of 4.0 ng/ml, and were TN”

 9.      Comment:  this should be added to the material and methods for patients: follow-up years etc for all groups. Are the authors comparing samples from patients with huge discrepancy in follow-up? This should also be explained

10.    Table 3 should be shown graphically (ROC curve) and number of patients should be consistent with the previous tables

11.    Figure 2 should show the distribution of NEM and PSA in the different groups and they should be compared statistically.

12.    Figure 3 should show the 2 ROC curves in one figure (2 colors), the 2 cohorts should have the same size, and add to the legend

Manuscript:

“NEM and tumor progression: Currently, there is no blood-based biomarker that can reliably predict the tumor stage of PCa. This is still performed by invasive procedures such as needle biopsies followed by histologic evaluation[2]. We examined whether plasma NEM levels significantly increase with increase in tumor stage. The results of Table 5 show that plasma NEM levels increased with increase in tumor stage, but plasma PSA levels did not.”

13.    Comment: Figure 3 should show the 2 ROC curves in one figure (2 colors), the 2 cohorts should have the same size, and add to the legend. There is no Table 4, 5 and 6 in the manuscript. Anyway, these data should be shown in a Figure. Table 4 is not mentioned in the text.

Manuscript:

“Changes in serum NEM levels post-treatment patients: Samples from post-prostatectomy patients were divided into three groups: 1) patients with the evidence of BCR; 2) thoseCancers 2023, 15, x FOR PEER REVIEW 8 of 11 with no signs of BCR; and 3) patients with distant metastases. Table 6 show that both PSA and NEM showed significant, many-fold increase their levels in patients with BCR as compared to those without BCR. Similarly, many fold increases were also observed in patients with metastatic cancer.

Serum NEM levels in other cancers: We examined 80 samples from a variety of other cancer patients ranging from hematological malignancies to solid tumors. Generally, plasma NEM levels of these patients were higher than normal. However, the increase was moderate (11.87 ± 1.307; n=58), and the levels never reached those observed in most cases of PCa. Moreover, we could not identify a specific pattern for a certain type of cancer.”

 14.    Comment: The rest of the results are missing from the manuscript so I cannot comment.

Manuscript:

“To examine whether NEM can serve as a biomarker for post-treatment monitoring, we examined serum NEM levels of PCa patients who had undergone radical prostatectomy. Our cohort included subjects who had shown or did not show biochemical relapse (BCR) including changes in serum PSA levels”

 15.    Comment:

Authors should describe the time points and follow-up strategy for each patient group in materials and methods.

 Manuscript:

 “the results showed that the patients with BCR displayed several fold higher serum NEM levels than those that did not show BCR.”

 16.    Comment: this is missing from the manuscript

Manuscript:

“The first limitation of this study is the presence of predominant Caucasian population as compared to African Americans”

17.    Comment: this is also missing in materials and methods, no data are presented about ethnicity

Manuscript:

“Only NEM and PSA were used as predictive variables, which leaves open the possibility that the inclusion of other biomarker(s) may increase the reliability of testing.”

 18.  Comment: authors show a performance of their test as “the diagnosis of cancer has 98% sensitivity and 96% specificity.”, therefore this limitation does not really apply as the data are extremely good, this seems more like a diagnostic test rather than a screening test. One clear limitation is a missing validation cohort to confirm the predictive power of the biomarker. Another limitation is the collection of samples from a single clinic. Additionally, in the abstract they write “with sensitivity of 98% and specificity of 97”, please correct.

Manuscript:

“In summary, NEM was significantly a better predictor than PSA alone for PCa in elderly males.”

19.    Comment: I would remove “elderly”.

  20.    Comment: the abstract is missing the part about materials and methods. The results are far too extended and difficult to understand without a proper explanation of the study design. The wording should be improved.

 Reviewer conclusion: the manuscript needs major work to improve the way the data are shown and to better explain the study design, the work done and the results.

The performance of the test is very interesting, I hope the authors can improve the manuscript.

Author Response

Response to Reviewer 2:

We are extremely grateful to Reviewer 2 for the through review and excellent suggestions that will significantly improve our manuscript.

Point-to-point response to the Reviewer’s comments are presented as follows:

  1. Comment: Add space (in yellow):

This error was introduced by the bibliography program, which we did not notice. This has now been corrected in the revised MS.

  1. Comment: The reference to this assertion is an abstract where no direct comparison to an ELISA is performed. The author should specify what they mean with “at least 100X sensitive as ELISA” and adapt the claim.

The reference (15)  of our assay development publication has now been cited in the Methods section. Interassay Variation has been calculated for the present study and has been reported (please see page 5, lines 5-7).

  1. Comment: The authors should specify at which time points of the disease course the sample were collected. For prostate cancer patients: prior to biopsy, or prior to prostatectomy? During treatment? Which treatment?  For BPH patients: before, during or after treatment? For BCR patients: before prostatectomy? During BCR?

We have now added several amendments and new additions to address the issues raised by the Reviewer. We have tried to explain the goal of the study and study design  better than the first version.  We also added more details on the source of samples and their ethnic distribution. We have removed post-therapeutic part of the study as we did not have complete follow-up details of all patients. This will be examined in a follow up study with a proper study designed for that purpose. I hope the reviewer will be satisfied with our response (See page 4, Methods section; patients subsection)

  1. Comment: The authors should show these data or add the reference to a published manuscript.

This is about the assay details. We have provided the reference in the Methods section on Page 5 (Alzghoul et al, reference 15 in bibliography)

  1. Comment: Table 1A: age should show min-max, average and statistical comparison among the 4 groups.

We have taken the reviewer’s suggestion and revised Table 1. Please see Table 1C in the revised manuscript.

  1. Comment: Figure 1 is unclear. Number and groups of patients analysed is not clear. Authors should have shown the Pearson correlation with each single variable. What is the correlation to Cancer? Yes/no value? What is the purpose to correlate a continuos variable to a binary variable?It would have been much more valuable to show and compare the distribution of NEM in the 4 patient groups (if samples were taken at the same time point), for example by kruskal wallis test and box plot.

We apologize for this confusion. We have now corrected this by reorganizing the data in a single spreadsheet and then reanalyzing the data. We have also provided the explanation for inconsistency. Because the samples were sourced from three different institutions, the data that came with the samples was also different. Some sources provide complete patient details, others less than complete. However, the primary goal was to compare the efficacy of NEM in diagnosis of PCa, and we had enough data from all sources to evaluate this aspect of the study. Therefore, we have removed the data on post-treatment (BCR vs non-BCR) for a new study with an appropriate design. 

We have also taken the reviewer’s suggestion and the data is presented as box plot with statistical analysis between the four groups. Please see Revised Figure 1.

  1. Comment: Table 2A is missing something. Are the first 3 columns based on NEM and the second 3 on PSA as cut-off? This is missing. The cut off values in the text are different from those in the table anyway (1.8 instead of 1.787 and 4.26 and 4.216). These data would be best understood graphically with a box plot. The number of patients in the table do not correspond to table 1. If some patients have partial data they should be removed from the analysis and the different groups should be consistent in size. Same comments for Table 2B and C.

We have taken the Reviewer’s suggestion, made sure all samples are included in the analysis and the data is now presented in Figure 1 of the revised MS.

  1. Comment:  I would suggest to use the data to generate a ROC curve and then detect the cut-off at a specific sensitivity.

We have accepted the Reviewer’s suggestion, and derived cut-off values from ROC curves. Please see Figure 2 and Table 2 (A-D) in the revised MS.

  1. Comment:  this should be added to the material and methods for patients: follow-up years etc for all groups. Are the authors comparing samples from patients with huge discrepancy in follow-up? This should also be explained

Since we have removed the data on post-therapeutic follow up, this comment has been addressed.

  1. Table 3 should be shown graphically (ROC curve) and number of patients should be consistent with the previous tables

We have derived cut-off values from ROC curve, and then examined whether the biomarker levels for each patient are consistent with their clinical/pathological data. The patient numbers in each category are now consistent. Please see Figure 2 and Tables 2 in the revised MS.

  1. Figure 2 should show the distribution of NEM and PSA in the different groups and they should be compared statistically.

We have already compared different groups statistically in Figure 1 of the revised MS. The goal of Figure 4 (in the revised MS, was figure 2 in our earlier MS) was to highlight the ability of NEM to discriminate benign from cancer when PSA is high in both groups. We did not explain this properly in our MS.  Hopefully, this will be more clear in the revised MS.

  1. Figure 3 should show the 2 ROC curves in one figure (2 colors), the 2 cohorts should have the same size, and add to the legend

It has been revised per Reviewer’s suggestions (Please see revised Figure 2 and Legend)

  1. Comment: Figure 3 should show the 2 ROC curves in one figure (2 colors), the 2 cohorts should have the same size, and add to the legend. There is no Table 4, 5 and 6 in the manuscript. Anyway, these data should be shown in a Figure. Table 4 is not mentioned in the text.

Please see response to Comment 12. I apologize for the omission, possibly happened while uploading the MS.

  1. Comment: The rest of the results are missing from the manuscript so I cannot comment.

The MS has been revised to ensure all data reported are discussed. Apology for the inadvertent omission.

  1. Comment:

Authors should describe the time points and follow-up strategy for each patient group in materials and methods.

The post-treatment part of the study has been removed because of the incomplete data.

  1. Comment: this is missing from the manuscript

Same as the response for Comment 15.

  1. Comment: this is also missing in materials and methods, no data are presented about ethnicity

This has now been provided in Table 1B as a part of Material and Methods

  1. Comment: authors show a performance of their test as “the diagnosis of cancer has 98% sensitivity and 96% specificity.”, therefore this limitation does not really apply as the data are extremely good, this seems more like a diagnostic test rather than a screening test. One clear limitation is a missing validation cohort to confirm the predictive power of the biomarker. Another limitation is the collection of samples from a single clinic. Additionally, in the abstract they write “with sensitivity of 98% and specificity of 97”, please correct.

I greatly appreciate these comments of Reviewer. We have revised this section after reading your comments. Please see last paragraph on Page 12 of the revised MS.

“In summary, NEM was significantly a better predictor than PSA alone for PCa in elderly males.”

  1. Comment: I would remove “elderly”.

The word “elderly” has been removed.

  1. Comment: the abstract is missing the part about materials and methods. The results are far too extended and difficult to understand without a proper explanation of the study design. The wording should be improved.

We have revised the abstract to address reviewer’s comments.

Round 2

Reviewer 2 Report

Comments and Suggestions for Authors

Most of my comments have been addressed. I have two major concern, the author write: «For PCa patients, the samples were collected after prostatectomy at intervals of several months apart, at the same time that PSA was sampled, showing either a recurrence or no recurrence. «

If I understand this correctly, samples for «PCa group» were collected at random time points after prostatectomy and, if positive for PSA (which should be around 0) they were considered PCa patients, if PSA was negative (no biochemical recurrence), the patients were considered negative for PCa. 

1. In my opinion, the «PCa group» should be renamed «BR+ PCa group» and the title and claims should be adapted accordingly. For example, the title should be «Predicting the biochemical recurrence of prostate cancer with a novel blood-based biomarker: comparison of its performance with prostate-specific antigen». Of note, is that PSA is used to determine the BR in a binary manner (negative or positive, not as a concentration), therefore in my opinion this analysis is flawed and this concept should be at least clearly highlighted in the introduction and discussion. Normally PSA is used to select patients at risk of PCa, collecting samples before biopsy and prostatectomy, thus comparing samples (NEM) and PSA collected pre-biopsy to samples and PSA post-prostatectomy is should be addressed as a clear limitation of this study.

2. Why were the samples collected post-prostatectomy and BR- not part of the analysis? This would have been a good control group for the prediction of BR+ PCa. I suggest the author to add this data.

3. Minor language mistakes are present in the text and should be corrected. For example insetad of «the corresponding numbers” it should be “the corresponding concentrations”.

Comments on the Quality of English Language

Minor corrections needed

Author Response

Most of my comments have been addressed. I have two major concern, the author write: «For PCa patients, the samples were collected after prostatectomy at intervals of several months apart, at the same time that PSA was sampled, showing either a recurrence or no recurrence. «

If I understand this correctly, samples for «PCa group» were collected at random time points after prostatectomy and, if positive for PSA (which should be around 0) they were considered PCa patients, if PSA was negative (no biochemical recurrence), the patients were considered negative for PCa. 

  1. In my opinion, the «PCa group» should be renamed «BR+ PCa group» and the title and claims should be adapted accordingly. For example, the title should be «Predicting the biochemical recurrence of prostate cancer with a novel blood-based biomarker: comparison of its performance with prostate-specific antigen». Of note, is that PSA is used to determine the BR in a binary manner (negative or positive, not as a concentration), therefore in my opinion this analysis is flawed and this concept should be at least clearly highlighted in the introduction and discussion. Normally PSA is used to select patients at risk of PCa, collecting samples before biopsy and prostatectomy, thus comparing samples (NEM) and PSA collected pre-biopsy to samples and PSA post-prostatectomy is should be addressed as a clear limitation of this study.

We have deleted the sentences of concern which unfortunately caused a misunderstanding on the part of the reviewer.  The PCa group is the group that had plasma samples and was subsequently histologically confirmed by biopsy to have prostate cancer.  There is no collection after prostatectomy in this dataset. There were 48 patients who did have plasma collected after prostatectomy (which would contain the BR+ group).  However, since we lacked sufficient follow-up on these cases, we have removed them from the Materials and Methods section of the manuscript. We also removed a small extraneous Table pertaining to these 48 men.

The words “subsequent histologic” were added to the Abstract to clarify that the plasma measurements preceded histologic diagnosis.

  1. Why were the samples collected post-prostatectomy and BR- not part of the analysis? This would have been a good control group for the prediction of BR+ PCa. I suggest the author to add this data.

 Again, we are no longer attempting to include these 48 patients in this study and this would be a project for future study.

  1. Minor language mistakes are present in the text and should be corrected. For example insetad of «the corresponding numbers” it should be “the corresponding concentrations”.

This has been changed in the Abstract.  Also, minor grammatical or word choice improvements have been made here and there throughout the text.
